# Altered hepatic metabolism mediates sepsis preventive effects of reduced glucose supply in infected preterm newborns

Ole Bæk[1,2†], Tik Muk[1†], Ziyuan Wu[1], Yongxin Ye[1,3], Bekzod Khakimov[3], Alessandra Maria Casano[4], Bagirath Gangadharan[4‡], Ivan Bilic[4], Anders Brunse[1], Per Torp Sangild[1,2], Duc Ninh Nguyen[1]*

[1]Section for Comparative Pediatrics and Nutrition, Department of Veterinary and Animal Sciences, University of Copenhagen, Copenhagen, Denmark; [2]Department of Neonatology, Rigshospitalet, Copenhagen, Denmark; [3]Department of Food Science, University of Copenhagen, Copenhagen, Denmark; [4]Plasma-derived therapies, Baxalta Innovations GmbH, part of Takeda Pharmaceuticals Ltd, Vienna, Austria

*For correspondence: dnn@sund.ku.dk

†These authors contributed equally to this work

Present address: ‡independent consultant, Vienna, Austria

## eLife Assessment

The study follows up on previous work suggesting that lower glucose concentrations are protective from sepsis but put the patient at risk for hypoglycemia. In this paper, the authors identify that a slightly higher dose of glucose is still protective but no longer puts the patients at risk for hypoglycemia. The study is **important**, supported by **convincing** data, and will be of interest to a broad audience.

**Abstract** Preterm infants are susceptible to neonatal sepsis, a syndrome of pro-inflammatory activity, organ damage, and altered metabolism following infection. Given the unique metabolic challenges and poor glucose regulatory capacity of preterm infants, their glucose intake during infection may have a high impact on the degree of metabolism dysregulation and organ damage. Using a preterm pig model of neonatal sepsis, we previously showed that a drastic restriction in glucose supply during infection protects against sepsis via suppression of glycolysis-induced inflammation, but results in severe hypoglycemia. Now we explored clinically relevant options for reducing glucose intake to decrease sepsis risk, without causing hypoglycemia and further explore the involvement of the liver in these protective effects. We found that a reduced glucose regime during infection increased survival via reduced pro-inflammatory response, while maintaining normoglycemia. Mechanistically, this intervention enhanced hepatic oxidative phosphorylation and possibly gluconeogenesis, and dampened both circulating and hepatic inflammation. However, switching from a high to a reduced glucose supply after the debut of clinical symptoms did not prevent sepsis, suggesting metabolic conditions at the start of infection are key in driving the outcome. Finally, an early therapy with purified human inter-alpha inhibitor protein, a liver-derived anti-inflammatory protein, partially reversed the effects of low parenteral glucose provision, likely by inhibiting neutrophil functions that mediate pathogen clearance. Our findings suggest a clinically relevant regime of reduced glucose supply for infected preterm infants could prevent or delay the development of sepsis in vulnerable neonates.

**eLife digest** Newborn babies, especially those born prematurely, are highly vulnerable to infections. This can then lead to sepsis, a life-threatening condition where the immune system overreacts to an infection and damages organs. Immune cells are fueled by nutrients, especially glucose (sugar), when they help defend the body against dangerous microorganisms. Therefore, high blood glucose levels can heighten the risk that these infection responses become too powerful and harm the body. However, too little glucose can also result in low blood sugar levels which can cause brain damage. This represents a key challenge in neonatal care, especially for babies which cannot be fed by mouth and therefore require nutrients to be delivered directly to their bloodstream (parenteral nutrition). Infants also struggle to regulate blood sugar levels on their own, which affects their response to infection.

The liver plays a major role in both immune responses and in converting nutrients into energy forms that our body can use. Adjusting glucose intake may influence how the liver generates energy and, in turn, prevent premature babies from developing sepsis.

To address this question, Bæk, Muk et al. infused premature piglets with parenteral nutrition containing different levels of glucose, starting shortly after birth. The health of the animals was then monitored before and after being infected with a bacterium *Staphylococcus epidermidis*, a common causative agent of infection in very preterm infants. The experiments showed that pigs that received lower levels of glucose were more likely to survive the infection than those that were given higher doses.

Despite receiving less glucose, the pigs managed to maintain their blood sugar levels within a safe range. This is because their livers compensated by shifting to using other sources of energy instead, and by manufacturing their own glucose. Further experiments found that adjusting glucose intake only protected against sepsis if the change was made early during infection, before the individual developed symptoms.

These findings suggest that adjusting the levels of glucose given to premature babies in hospital could be a simple, clinically relevant strategy to prevent sepsis. Further research is needed to explore how the liver uses alternative sources of energy to manufacture glucose and, in turn, to maintain normal blood sugar levels in infants.

## Introduction

Preterm infants (born before 37 wk of gestation) are particularly susceptible to serious neonatal infections potentially leading to sepsis, a state of life-threatening organ dysfunction (*Hibbert et al., 2018*). Most episodes of late-onset sepsis in these infants (occurring >48 hr after birth) are caused by coagulase-negative staphylococci (CONS), such as *Staphylococcus epidermidis* (*Strunk et al., 2021*; *Shane et al., 2017*), which may enter the circulation via indwelling medical devices or from the gastrointestinal tract (*Deitch, 2012*; *Dong et al., 2018*). The development of sepsis is often attributed to the immature immune systems in preterm infants, but a recent concept of immunometabolism suggests it may partly be caused by the distinct newborn metabolic state, affected by their limited energy reservoirs (*Harbeson et al., 2018*). Upon exogenous challenge, immune cells and many other cell types in healthy adults undergo a metabolic shift from oxidative phosphorylation (OXPHOS) to glycolysis, producing large amounts of adenosine triphosphate (ATP) to fuel inflammatory responses, facilitating efficient reduction of pathogen burden through immunological resistance (*Harbeson et al., 2018*; *Gaber et al., 2017*). However, immune cells can also employ a tolerance strategy, limiting the active resistance response and associated immunopathology, via an opposite shift from glycolysis to OXPHOS (*Harbeson et al., 2018*). This tolerance strategy is found in newborns, especially those born prematurely, as they have low energy stores, but high demands for growth and development. Therefore, energy tends to be prioritized for vital organ functions rather than immune responses to external stimuli, explaining why they display low immune cell metabolism (*Harbeson et al., 2018*; *Kollmann et al., 2012*) and can withstand 10–100 times greater pathogen loads than adults before showing clinical symptoms (*Strnad et al., 2017*).

The liver plays a critical role during bloodstream infections, being a well-perfused organ with resident immune cells. Besides its role in producing important immunomodulatory plasma proteins,

the role of hepatic metabolism during infections has recently emerged (*Strnad et al., 2017*). For instance, blocking hepatic glycolysis lowers overall energy expenditure and improves OXPHOS, which facilitates immune tolerance and better survival during polymicrobial sepsis in mice (*Mainali et al., 2021*). This suggests that manipulation of hepatic energy metabolism may affect the systemic immune responses and be a promising avenue to reduce morbidity during neonatal infections. Progression of sepsis is also concomitant with the activation of neutrophils, may lead to increased activity of neutrophil-derived serine proteases that play a key role in sepsis-related tissue damage (*Burg and Pillinger, 2001*). Their inhibition has likewise been proposed as a therapeutic strategy to resolve septic conditions (*Lord et al., 2020*). Recent studies have shown that the components of one of the most abundant plasma protein complexes, Inter-Alpha Inhibitor Protein (IAIP), are differentially regulated during sepsis leading to a significant reduction of the whole complex both in adults (*Lim et al., 2003*; *Opal et al., 2007*; *Rucevic et al., 2007*) and neonates (*Baek et al., 2003*; *Chaaban et al., 2009*). Administration of enriched human plasma-derived IAIP (hIAIP) in a murine model of neonatal sepsis has been shown to reduce disease severity and mortality through suppression of pro-inflammatory responses (*Singh et al., 2010*).

Due to the lack of mother's milk and immature gut, preterm infants often receive glucose-rich parenteral nutrition (PN) during the first few weeks of life to ensure sufficient nutrition until full enteral nutrition can be established. The glucose levels are tailored to provide energy for growth and to avoid hypoglycemia-induced brain injury (*Mesotten et al., 2018*; *McGuire et al., 2004*; *Alaedeen et al., 2006*; *Harris et al., 2012*). However, this practice leads to hyperglycemia in up to 80% of very preterm infants during early life (*Beardsall et al., 2010*). Prolonged glucose-rich PN is related to longer hospitalization in septic infants (*Alaedeen et al., 2006*), which suggests it may affect the blood immune-metabolic axis or hepatic metabolism during infection, facilitating glycolysis and active immune resistance. This is in line with a recent study showing that limited nutrition in the acute phase of bacterial infection in rodents reduced glycolysis and enhanced disease tolerance, leading to better survival (*Wang et al., 2016*). Of note, there are no specific guidelines for parenteral glucose intake during serious neonatal infections.

Preterm pigs provide a unique, prematurely born, animal model, enabling PN via an umbilical catheter and mimicking clinical and cellular infection responses to *S. epidermidis* (*Bæk et al., 2020b*; *Brunse et al., 2018*), as seen in septic preterm infants similarly infected with CONS (*Strunk et al., 2021*). With this model, we have previously shown that the high parenteral glucose supply used in neonatal intensive care units exaggerated systemic glycolysis, triggering hyper-inflammatory responses and clinical sepsis signs, whereas strict glucose restriction prevented sepsis, but caused hypoglycemia (*Muk et al., 2022*). Here, we further exploited this model to explore how glucose regimes affect hepatic metabolism and glucose homeostasis to both prevent hypoglycemia and to alleviate the risk of sepsis in CONS infected preterm newborns. We identified a reduced parenteral glucose regimen that during neonatal infection both maintained normoglycemia and modulated the hepatic immune-metabolic networks in a manner that avoided excessive energy used for immune responses, leading to improved overall survival. However, postponing the reduction in glucose supply until clinical symptoms started to manifest did not affect clinical deterioration, suggesting metabolic constraints early in the immune response are critical for determining the fate. Lastly, to further explore potential optimizations of the treatment regimen, we evaluated concomitant treatment of preterm pigs with reduced PN glucose regimen supplemented with intravenous administration of purified hIAIP. In contrast to our original hypothesis, hIAIP supplementation reversed the effects of glucose reduction in this model of neonatal sepsis and provided no survival benefit.

Collectively, these clinically relevant findings suggest that reduced parenteral glucose supply, together with antibiotics and other supporting care, as a concomitant therapy for CONS infected preterm infants may prevent or delay the development of neonatal sepsis.

## Results

### Reduced glucose supply improves glucose homeostasis and survival during neonatal infection

Using preterm pigs as models for infected preterm infants, we have previously shown that hyperglycemia induced by a high parenteral glucose supply (High, 21%, 30 g/kg/d), is predisposed to sepsis

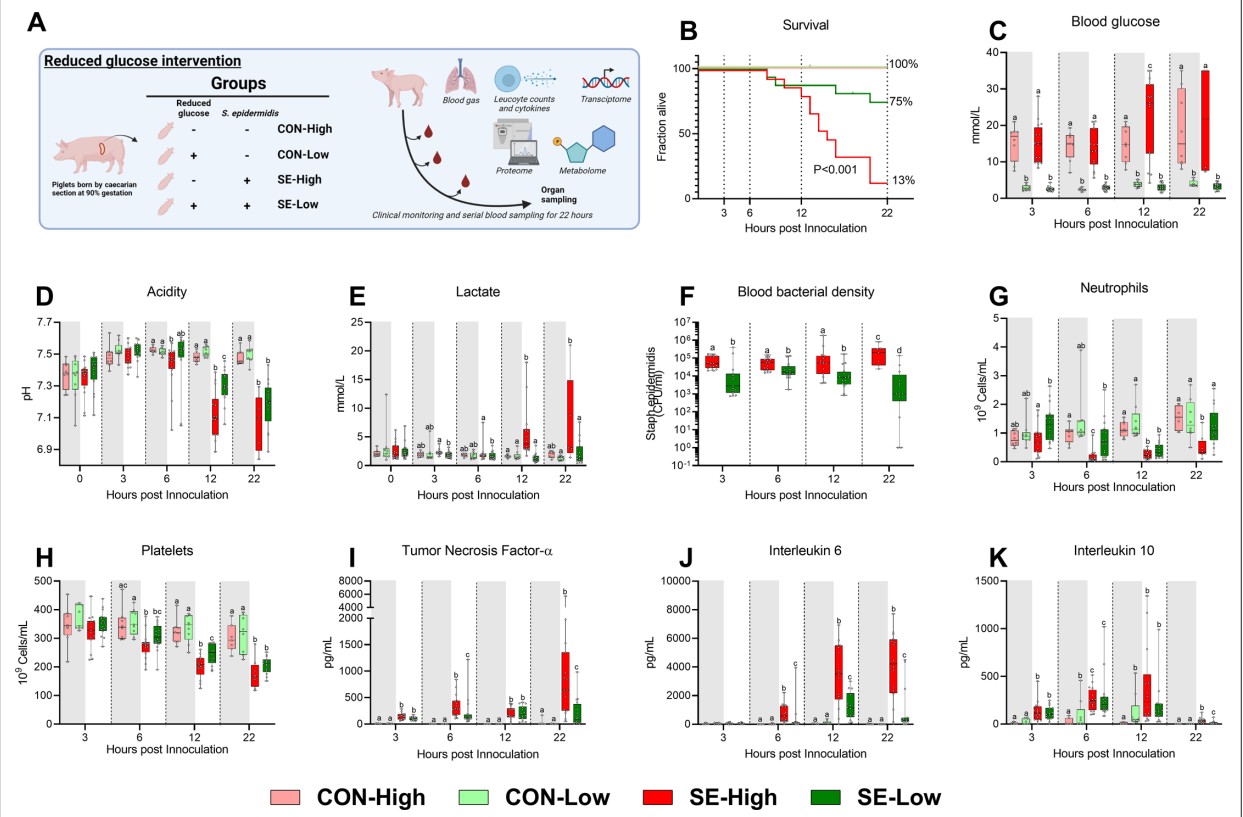

**Figure 1.** Animal study overview, clinical, and immunological results. (**A**) Study overview, preterm pigs were nourished with parenteral nutrition (PN) with either high (21%, 30 g/kg/d) or low (5%, 7.2 g/kg/d) glucose supply and infused with either *Staphylococcus epidermidis* or control saline. Animals were followed for 22 hr and blood samples were collected for further analysis. This panel was created with BioRender.com. (**B**) Survival of animals during an experiment, presented as time to euthanasia according to predefined humane endpoints with corresponding log-rank test comparing *S. epidermidis* (SE)-High and SE-Low (**C**) Blood glucose, measured by glucose meter 3, 6, and 12 hr after SE inoculation as well as at euthanasia, presented as 95% box plots. (**D, E**) Blood gas parameters collected before inoculation with SE (0 hr) and 3, 6, and 12 hr after as well as at euthanasia, presented as 95% box plots. (**F**) Blood bacterial density in infected groups at 3, 6, 12, and 22 hr after bacterial inoculation, presented as 95% box plots on a logarithmic scale. (**G–K**) Blood hematology and plasma cytokines at 3, 6, 12 and 22 hr after bacterial inoculation, presented as 95% box plots. (**C–K**) Data at each time point analyzed separately, bars labeled with different letters are significantly different from each other (p<0.05), n=8–9 for control animals, and 10–16 for infected groups.

The online version of this article includes the following figure supplement(s) for figure 1:

**Figure supplement 1.** Supplemental blood gas, bacterial density and hematological parameters.

and high mortality following *S. epidermidis* infection. However, a very restrictive glucose supply, below the recommended guidelines for preterm infants (*Mesotten et al., 2018*) (1.4%, 2 g/kg/d) protected against the development of sepsis, but caused severe hypoglycaemia (*Muk et al., 2022*).

Here, we investigated whether it was feasible to reduce parenteral glucose supply during neonatal infection to an extent that reduced the severity of symptoms, while avoiding hypoglycemia. Preterm pigs were delivered by caesarian section at 90% gestation and nourished with PN using either the high or a reduced glucose (Low, 5%, 7.2 g/kg/d) regimen (*Mesotten et al., 2018*). The animals were systemically infused with either live *S. epidermidis* (SE, $10^9$ colony forming units (CFU)/kg) or control saline (CON), (see *Figure 1A* for study overview). Animals were then closely monitored for 22 hr, and those showing severe septic symptoms (deep lethargy, hypoperfusion, and blood pH <7.1 as signs of severe acidosis and impending respiratory/circulatory collapse) were euthanized according to predefined humane endpoints.

Reduced glucose supply markedly improved survival following infection, with an almost six-fold reduction in the absolute risk of death (p<0.001, *Figure 1B*). The reduced glucose supply expectedly led to lower blood glucose levels, compared to SE-High animals, and blood glucose levels remained stable around 2–3 mmol/l during the whole experiment (*Figure 1C*). Blood pH started to decrease

at a slower rate from 6 hr onwards in SE-Low animals with correspondingly lower lactate levels, indicating slower rates of glycolytic breakdown of pyruvate (*Figure 1D and E*). Of note, base excess was improved in SE-Low animals already from 3 hr after the start of the infection, whereas glucose supply during infection had no impact on partial carbon dioxide pressure (*Figure 1—figure supplement 1A and B*). This indicates that the severity of the metabolic acidosis occurring during neonatal sepsis was alleviated by the reduced glucose regimen. Importantly, despite the marked difference in blood glucose levels, blood osmolality did not differ much among the groups (*Figure 1—figure supplement 1C*), underlining that the differences in clinical responses were not driven by a hyperosmolar state induced by the high glucose supply. As such, it appeared that reduction in parenteral glucose supply improved survival and clinical symptoms while still maintaining normoglycemia, in the lower range.

## Glucose homeostasis affects cellular responses to neonatal infection

With a clear clinical benefit of reduced glucose supply observed, we then further investigated the potential underlying mechanisms by measuring the immune cell and molecular responses to the infection. SE-Low animals showed consistently lower blood bacterial densities for the duration of the experiment (*Figure 1F*). All the main blood leucocyte subsets dropped in response to the infection (*Figure 1—figure supplement 1D, E*), but depletions in circulating neutrophils were more pronounced in SE-High vs. SE-Low animals, evident at 6 h post-infection which were only replenished in SE-Low animals (*Figure 1G*), suggesting either better bone marrow capacity or less neutrophil activation, although this was not further investigated. Also, SE-Low animals showed less severe thrombocytopenia (*Figure 1H*), and depletions of these blood cellular subsets are known to be associated with increased severity in neonatal sepsis (*Melvan et al., 2010; Panda et al., 2021; Timperi et al., 2016*).

In parallel, it was evident that the inflammatory response in infected animals with a reduced glucose supply also was dramatically attenuated, with lower levels of plasma TNF-α, IL-6, and IL-10, compared to animals with a high glucose supply (*Figure 1I–K*). Differences in TNF-α and IL-6 levels were already evident at 6 hr after inoculation. Interestingly, among uninfected control animals, a reduced glucose supply also increased IL-10 levels (*Figure 1K*).

This indicates that a reduced parenteral glucose supply protected against neonatal sepsis via decreased bacterial burden and dampened pro-inflammatory immune responses. Relative to our previous findings, we showed that a more moderate reduction in glucose intake provided similar positive benefits in outcomes, while avoiding hypoglycemia (*Muk et al., 2022*).

## Reduced glucose supply increases hepatic OXPHOS and gluconeogenesis and attenuates inflammatory pathways

Given that both glucose levels and bacterial burdens were lower in the reduced glucose group we speculated that proliferation of bacteria was driving the phenotype. However, within SE-Low and SE-High pigs, mortality did not correlate to bacterial burdens (*Figure 1—figure supplement 1*). Although, we cannot exclude that circulating glucose levels affected bacterial proliferation, we hypothesized that an important mechanism behind these positive effects was a reduction of hepatic and circulating glycolytic metabolism, which may lead to rewiring immune cells to rely on other substrates than glucose, thereby constraining the overall pro-inflammatory cascades. Thus, we explored how glucose supply affected immune-metabolic gene expression in the liver, collected at euthanasia. Both infection and glucose supply dramatically impacted hepatic gene transcription, with more than 4000 differentially expressed genes (DEGs) for each group comparison (*Figure 2A*). Relative to SE-High, the SE-Low animals showed elevated pathways related to glycolysis/gluconeogenesis, OXPHOS, and TCA cycle, as well as fatty acid and glucogenic amino acid metabolism (*Figure 2B,C, Figure 2—figure supplement 1* and all DEGs can be found in *Figure 2—source data 1*). Though glycolysis and gluconeogenesis share many enzymes catalyzing reversible reactions, enzymes related solely to glycolysis or glycogen formation, such as glucokinase (*GCK*) and phosphoglucomutase 2 (*PGM2*) were downregulated in SE-Low compared to SE-High (*Figure 3D and E*), whereas those only involved in gluconeogenesis, such as fructose-bisphosphatase 2 (*FBP2*) and pyruvate carboxylase (*PC*) were upregulated (*Figure 3F and G*). This suggests that a shift from glycolysis to gluconeogenesis occurs in infected animals kept on a reduced glucose supply, even during infections. In parallel, SE-Low animals showed multiple attenuated pathways related to inflammation, including NF-kappa B, IL-17, TNF, and Transforming growth factor-β (TGF-β) signaling (*Figure 2C, Figure 2—figure supplement 1*). Within

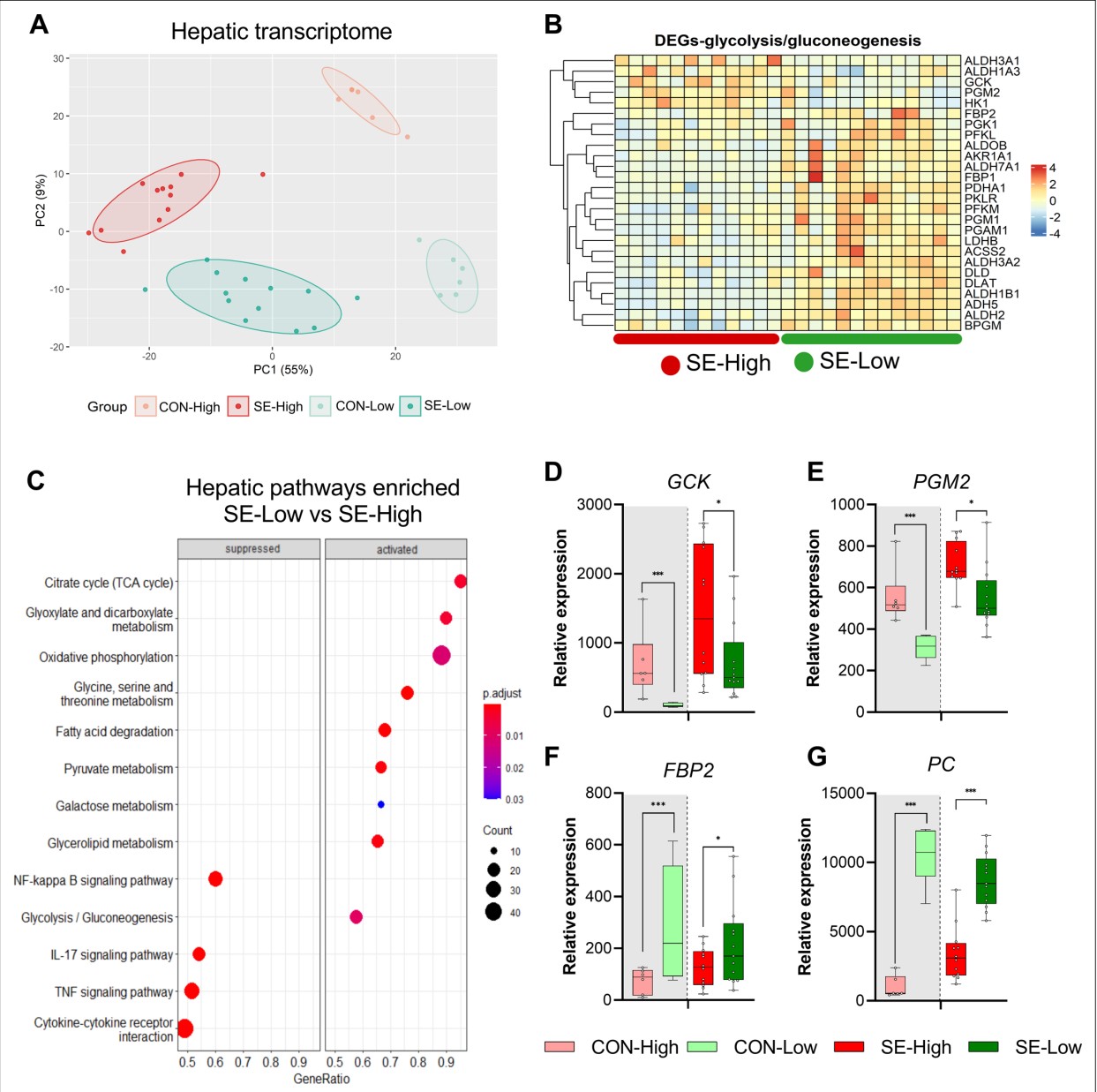

**Figure 2.** Impacts of glucose supply on liver transcriptomics at euthanasia. (**A**) Principal component analysis plot of transcriptomic profiles among the four groups. (**B**) Heatmap of differentially expressed genes (DEGs) related to glycolysis/gluconeogenesis, in the enriched pathways between the two infected groups. Differences shown as Z-scores, where red color indicates a higher expression and blue a lower. (**C**) Gene set enrichment analysis (GSEA) with gene ontology database showing the top pathways activated and suppressed by reduced glucose supply in infected animals. Size of the dots indicates the number of DEGs while the red color indicates a lower adjusted p-value. (**D–G**) Expression of genes exclusively related to glycolysis or gluconeogenesis, shown as relative expressions using 95% box plots. *p<0.05, ***p<0.001, n=7–8 for each control group and 15–16 for infected groups.

The online version of this article includes the following source data and figure supplement(s) for figure 2:

**Source data 1.** Hepatic transcriptome data.

**Figure supplement 1.** Metabolic pathways in liver transcriptome affected by reduced glucose supply.

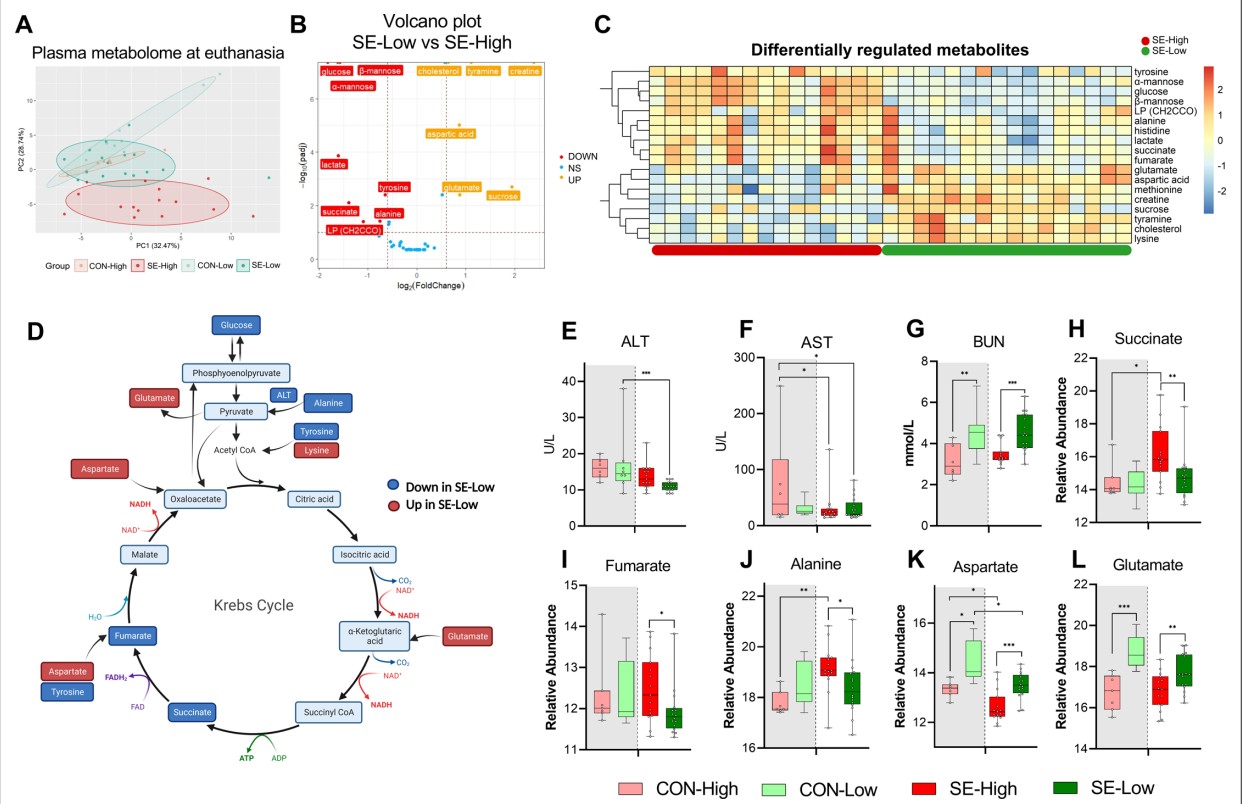

**Figure 3.** Impacts of glucose supply on plasma metabolism response revealed by proton nuclear magnetic resonance (NMR)-based metabolomics at euthanasia, either at the humane endpoint or 22 hr after bacterial inoculation. (**A**) Score plot of principal component analysis performed on metabolomics data among the four groups. (**B**) Volcano plot showing the differential expressed metabolites (DEMs) in *S. epidermidis* (SE)-Low vs SE-High. Yellow color indicates higher plasma levels, red lower. (**C**) Heatmaps of identified DEMs between the two infected groups. Differences shown as Z-scores, where red color indicates a higher expression and blue a lower. (**D**) Schematic representation of gluconeogenesis and TCA cycle metabolites altered between the two infected groups. Red represents a metabolite upregulated in SE-Low vs SE-High whereas dark blue indicates a metabolite downregulated; light blue indicates a metabolite not detected by [1]H NMR. (**E–G**) Plasma alanine transaminase (ALT), aspartate transaminase (AST), and BUN levels, shown as 95% box plots. (**H–L**) DEMs involved in gluconeogenesis and TCA cycle, shown as 95% box plots. *p<0.05, **p<0.01, ***p<0.001, n=7–8 for each control group and 15–16 for infected groups.

The online version of this article includes the following source data for figure 3:

**Source data 1.** Plasma metabolome data.

the two un-infected control groups, glucose supply affected similar metabolic processes, but not inflammatory pathways.

## Plasma metabolome confirms the glucose impact on gluconeogenesis and balanced TCA cycle

With the profound impact of infection and glucose supply on the transcribed hepatic metabolic pathways, we further sought to analyze the plasma metabolome at the time of euthanasia to gain more insights into these metabolic processes, especially to confirm if enhanced gluconeogenesis occurred in SE-Low animals. Via proton nuclear magnetic resonance ([1]H NMR) spectroscopy, we identified 70 major metabolite signals, among them, a total of 43 plasma metabolites were annotated. The metabolite profile of SE-High animals appeared to be distinct from the remaining three groups (*Figure 3A*). Infection changed abundance of 11 and 28 metabolites in reduced and high glucose conditions, respectively, while 35 metabolites were differentially regulated between the two infected groups (All differentially expressed metabolites shown in *Figure 3—source data 1*). Major annotated differentially regulated metabolites and blood biochemical parameters between SE-Low and SE-High groups were mainly sugars, amino acids, and TCA cycle intermediates (*Figure 3B and C*). *Figure 3D* highlights the detected metabolomic changes involved in the gluconeogenesis and TCA cycle. Importantly,

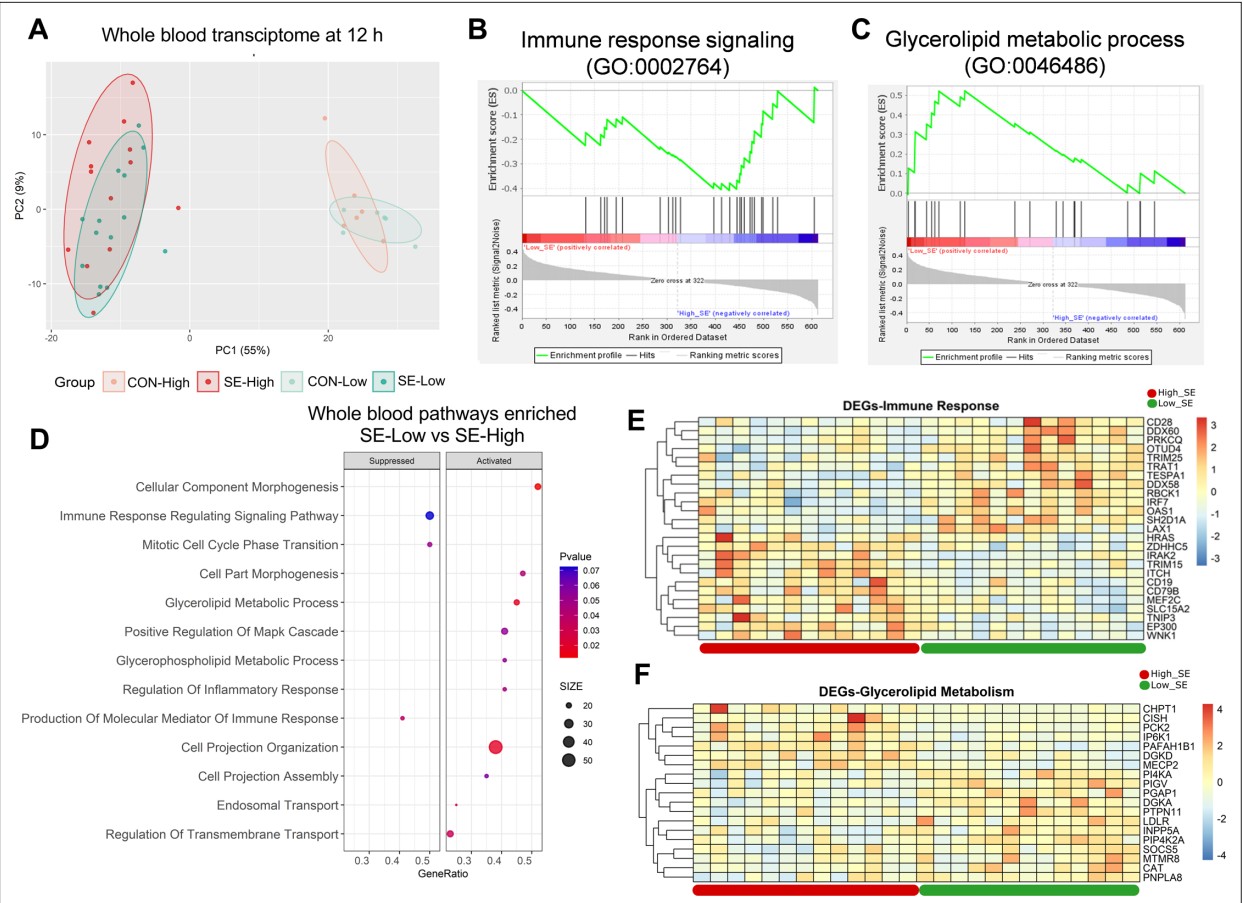

**Figure 4.** Impacts of glucose supply on whole blood transcriptomics at 12 hr after bacterial inoculation. (**A**) Score plot of principal component analysis of transcriptomic profiles performed among the four groups. (**B, C**) Gene set enrichment analysis (GSEA) of the immune response signaling (GO:0002764) with negative enrichment score and glycerolipid metabolic process (GO:0046486) with positive enrichment score in the *S. epidermidis* (SE)-Low, relative to SE-High animals. (**D**) Enrichment analyses using GSEA with gene ontology database showing the top pathways activated and suppressed by reduced glucose supply in infected animals. Size of dots indicates number of DEGs while red color indicates lower adjusted p-value. (**E, F**) Heatmaps of differentially expressed genes (DEGs) involved in the enriched immune response and glycerolipid metabolism pathways between the two infected groups. Differences shown as Z-scores, where red color indicates a higher expression and blue a lower. n=6 in each control group and 13–14 for infected groups.

The online version of this article includes the following source data and figure supplement(s) for figure 4:

**Source data 1.** Whole blood transriptome data.

**Figure supplement 1.** Inflammatory pathways in whole blood transcriptome affected by reduced glucose supply.

SE-High animals showed higher levels of succinate and fumarate (*Figure 3H–I*), further supporting that high glucose supply rewired systemic metabolism to heightened glycolysis, via inhibiting enzymes catalyzing succinate conversion, causing accumulation of succinate and fumarate (*Rosenberg et al., 2022*). On the other hand, SE-Low animals showed decreased plasma alanine and alanine aminotransferase activity, and increased levels of blood urea nitrogen, glutamate, and aspartate (*Figure 3E, G, J and L*), all pointing to the direction of enhanced gluconeogenesis by metabolizing glucogenic amino acids, possibly contributing to the maintenance of glucose homeostasis under reduced glucose supply.

## Only inflammatory but not metabolic network of blood transcriptome is modulated by reduced glucose supply

We then explored how infection and glucose supply affected blood leucocyte gene expressions at 12 hr post-infection challenge, the time-point when clinical outcomes started to drastically differ between the two infected groups. Infection itself dramatically altered the whole blood transcriptome

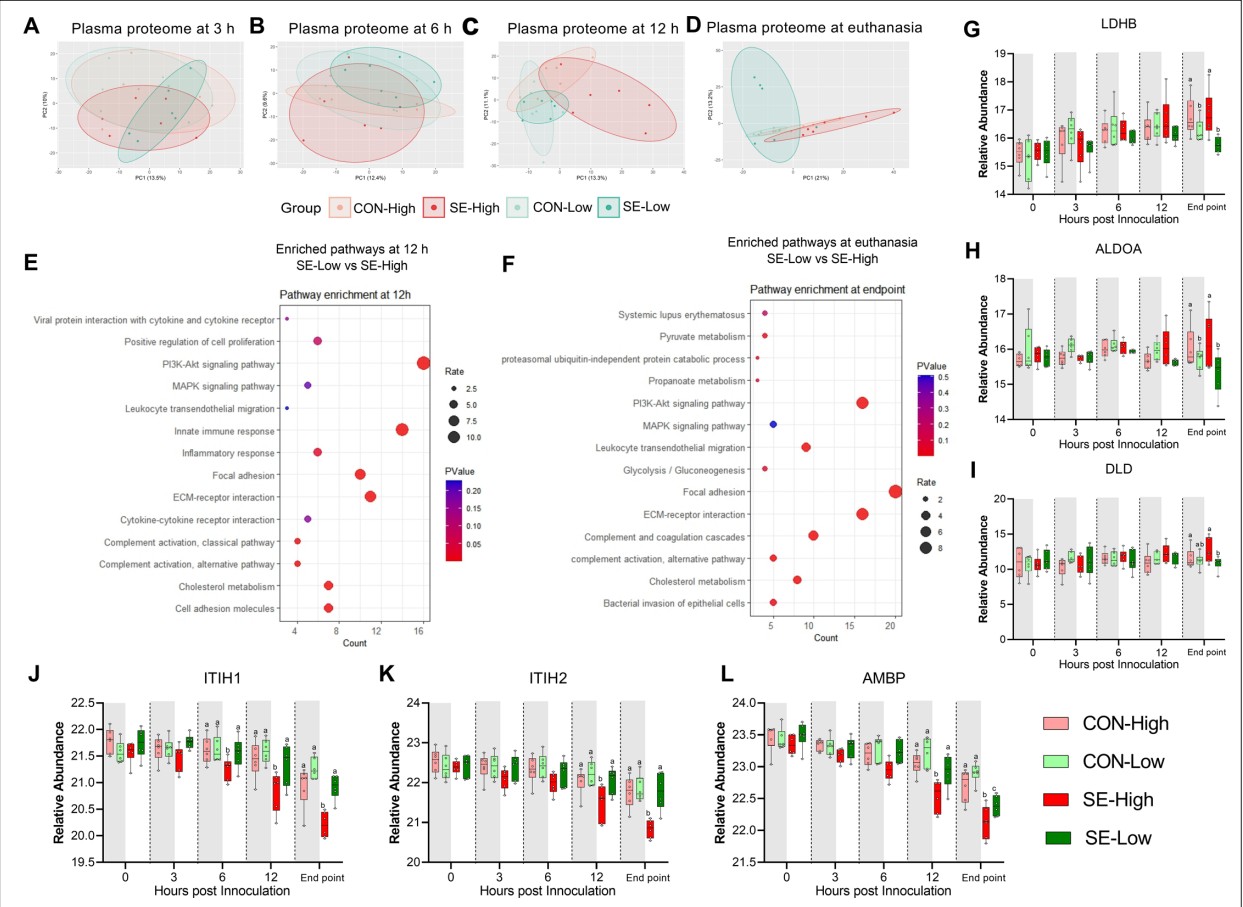

**Figure 5.** Impacts of glucose supply on plasma proteome at different time points post-bacterial infection. (**A–D**) Score plot of principal component analysis of proteome profiling changes from 3 hr to euthanasia among the four groups. (**E–F**) Top pathways differing between *S. epidermidis* (SE)-Low and SE-High at 12 hr post bacterial inoculation or euthanasia, were enriched using DAVID. Involved differential expressed proteins (DEPs) counts displayed on Xaxis and size of dots indicates number of DEPs, while red color indicates lower p-value. (**G–I**) DEPs involved in glycolysis or gluconeogenesis, including Lactate Dehydrogenase B (LDHB), Aldolase, Fructose-Bisphosphate A (ALDOA), Dihydrolipoamide dehydrogenase (DLD). (**J–L**) DEPs of porcine IAIP related proteins, heavy chains 1 and 2 (ITIH1, ITIH2) and Alpha-1-Microglobulin bikunin precursor (AMBP). (**G–L**) Shown as relative abundances with 95% box plots, data analyzed separately for each timepoint, bars labeled with different letters are significantly different from each other (p<0.05), n=6 in each group.

The online version of this article includes the following source data for figure 5:

**Source data 1.** Plasma proteome data.

(*Figure 4A*), with more than 5000 DEGs (all shown in *Figure 4—source data 1*). Glucose supply in un-infected conditions had limited impact, while within the infected groups differences in gene expression were much less pronounced than in the liver. The SE-Low animals showed 321 up-regulated and 363 down-regulated DEGs, compared to SE-High animals. Pathway enrichment analysis for DEGs between the two infected groups showed a dampened immune response, enhanced glycerolipid metabolic processes, and re-organization of cellular structure in SE-Low animals (*Figure 4B-E*, *Figure 4—figure supplement 1*). Surprisingly, no enriched pathways related to the metabolism of glucose, fatty acids, or amino acids were observed. These data strongly suggest that modulations of hepatic, but not circulating immune cell metabolism, by reduced glucose supply, play a central role in guiding systemic inflammatory responses in preterm neonates.

## Plasma proteome confirms groups of interacting mediators involved in immune-metabolic modulations

Clinical, transcriptome, and metabolome analysis at 12 hr post-infection and euthanasia clearly showed distinct immune-metabolic signatures between the two infected groups. To further capture

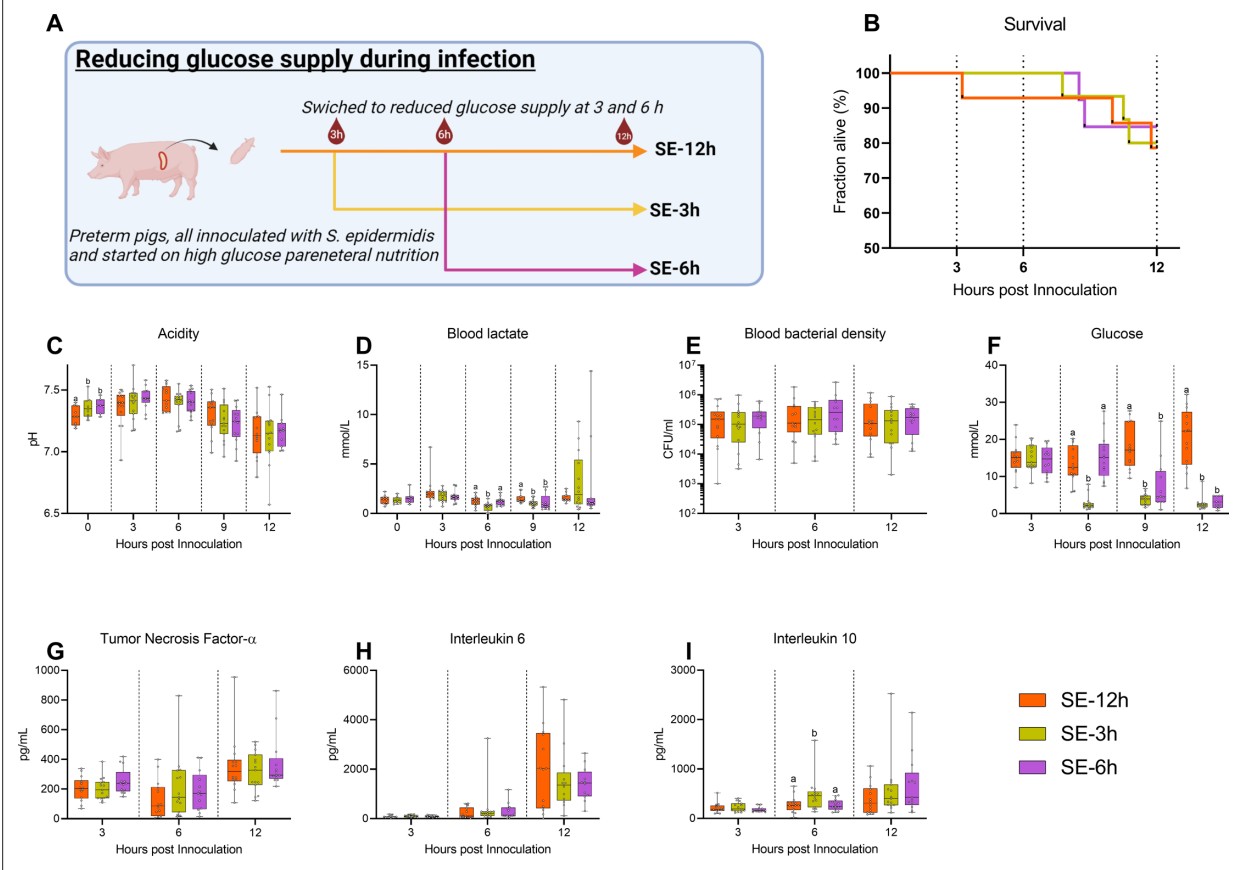

**Figure 6.** Follow-up experiment investigating effects of changing glucose regimen during infection. (**A**) Study overview, preterm pigs were all started on high glucose regimen and inoculated with *S. epidermidis*. After 3 hr one group was shifted to low glucose parenteral nutrition (PN) (SE-3 hr), while after 6 hr another group was similarly shifted (SE-6 hr) and the remaining pigs continued on high glucose PN for the rest of the experiment (SE-12 hr). This panel was created with BioRender.com. (**B**) Survival during experiment, presented as Kaplan-Meier curves. (**C, D**) Blood gas data collected at baseline and 3–12 hr after inoculation. (**E**) Blood bacterial density 3–12 hr after inoculation, shown on a logarithmic scale. (**F**) Blood glucose levels 3–12 hr after inoculation. (**G–I**) Plasma cytokine levels 3–12 hr after inoculation. (**C–I**) Presented as 95% box plots, data at each time point analyzed separately, bars labeled with different letters are significantly different from each other (p<0.05), n=11–14 for SE-12 hr, n=12–15 for SE-3 hr and n=10–12 for SE-6 hr.

The online version of this article includes the following source data and figure supplement(s) for figure 6:

**Source data 1.** Clinical and immunological data for follow-up experiment.

**Figure supplement 1.** Hematological parameters follwing reduction in glucose supply after 3 and 6 hours.

these differences, we performed plasma proteomics on longitudinal samples, from 3 hr post-bacterial challenge to euthanasia. Principal component analysis of proteomic profiles (*Figure 5A–D*) demonstrated no separation among the four groups at 3 hr and marginal separation of SE-High from SE-Low at 6 hr. The full impact of infection and glucose supply during infection was only obvious at 12 hr and euthanasia. These findings are in line with hematological and plasma cytokine data, which revealed only limited changes at 3–6 hr and more dramatic differences from 12 hr until euthanasia.

Due to relatively low DEPs between the two infected groups (93 and 179 proteins at 12 hr and euthanasia, respectively), we performed pathway enrichment analysis with all DEPs (*Figure 5— source data 1*). We detected similar enriched pathways to other transcriptome/metabolome data at 12 hr and euthanasia: immune response, pyruvate metabolism, and glycolysis/gluconeogenesis (*Figure 5E–F*). Specifically, levels of proteins encoding for enzyme conversion from pyruvate to lactate (LDHB, DLD) and upstream conversion of glucose in the glycolysis pathway (ALDOA) were higher in animals receiving high vs. reduced glucose supply, irrespectively of infection status, again showing increased use of glycolysis (*Figure 6G–I*). To further explore differences in inflammatory pathways we investigated plasma levels of IAIP protein subunits, as these are known to be reduced in infants with bacterial sepsis (*Baek et al., 2003*). Although, at the end of the experiment,

AMBP levels were lower in SE-Low compared to the uninfected group (*Figure 5L*), levels of heavy chains 1 and 2 as well as an alpha-1-microglobin precursor to Bikunin (AMBP) were significantly higher in SE-Low compared to SE-High, which correlates with a less severe inflammatory response (*Figure 5K–L*).

## Timing of glucose supply guides inflammatory and clinical outcomes post-infection

In a clinical setting, any treatment of infection would not be initiated before clinical symptoms become evident, which in our model occurs 3–6 hr after bacterial inoculation. We, therefore, investigated whether an intervention to reduce the high parenteral glucose supply at either 3 or 6 hr post-inoculation could rescue the animals from sepsis. *S. epidermidis* infected preterm pigs were all started on a high glucose supply and then switched to a reduced glucose supply at either 3 hr (SE-3 hr) or 6 hr post-inoculation (SE –6 hr) or kept on high glucose PN (SE-12 hr) and followed until 12 hr post-infection (see *Figure 6A* for an experimental overview).

No difference in survival could be detected among these three infected groups until 12 hr (*Figure 6B*). Likewise, although blood pH gradually dropped, it was similar among the groups 3–12 hr post-infection (*Figure 6C*). However, shifting to a reduced glucose supply led to lower lactate levels at 6 hr post-infection in SE-3 hr pigs, while at 9 hr the lower levels were observed in both shifted groups (*Figure 6D*). Although there were no differences in blood bacterial density at any timepoint (*Figure 6E*), blood glucose levels were quickly affected by shifting to a reduced glucose regime (*Figure 6F*), while only minor effects on hematological parameters were observed (*Figure 6—figure supplement 1*), with similar plasma TNF-α and IL-6 responses (*Figure 6G and H*). Plasma IL-10 levels were transiently elevated in the SE-3 hr group 6 hr after inoculation, indicating that some increase in anti-inflammatory responses (*Figure 6I*). In summary, glucose reduction during infection in this animal model did not improve clinical status until 12 hr post-challenge. Therefore, it appears that high blood glucose levels at the time of or shortly after the infection, rather than at clinical manifestation, are a determining factor for the clinical fate of infected preterm pigs.

## Administration of hIAIP combined with low glucose PN does not improve septic conditions

Since reduced parenteral glucose regimen during neonatal infection led to improved survival, but not complete protection from sepsis, by modulating the hepatic immune-metabolic networks leading to reduced inflammation: We, therefore, wished to explore whether concomitant intravenous administration of purified hIAIP, a liver-derived protein, could provide additional treatment benefits. Animals were provided with previously reported doses of hIAIP (*Singh et al., 2010*) at one and 12 h and monitored until 22 hr post-infection (SE-IAIP; *Figure 7A*). Supplementation of hIAIP increased mortality from 12 hr onwards (*Figure 7B*), but induced only limited alterations in blood acidity or lactate levels (*Figure 7C and D*). However, blood bacterial density was higher by the end of the experiment in the SE-IAIP group compared to SE-Low, despite similar glucose levels (*Figure 7E and F*). Interestingly there was a small, but significant, increase in plasma levels of IL-10 3 hr post-infection (*Figure 7G*). However, plasma levels of IL-6 or TNF-α, as well as blood leucocyte counts did not differ between SE-Low and SE-IAIP animals, although the SE-IAIP group did not recover neutrophils to the same degree as SE-Low by the end of the experiment (*Figure 7—figure supplement 1*). Therefore, in contrast to the hypothesized beneficial complementary effects of hIAIP treatment and low PN glucose regimen, the combined treatment resulted in reduced blood bacterial clearance in vivo and decreased survival of the infected animals.

To further explore the underlying cause of this phenotype, we turned to an ex vivo approach, assessing phagocytosis of bacteria by cord blood neutrophils from preterm pigs, treated with increasing hIAIP concentrations. The percentage of phagocytic neutrophils and the number of engulfed bacteria were inversely proportional to the levels of hIAIP received (*Figure 7H–I*). This demonstrated an inhibitory effect on neutrophil activation in vitro, thus suggesting that hIAIP-mediated reduction in neutrophil phagocytic capacity results in increased pathogen load and reversal of effects induced by the reduced glucose regimen.

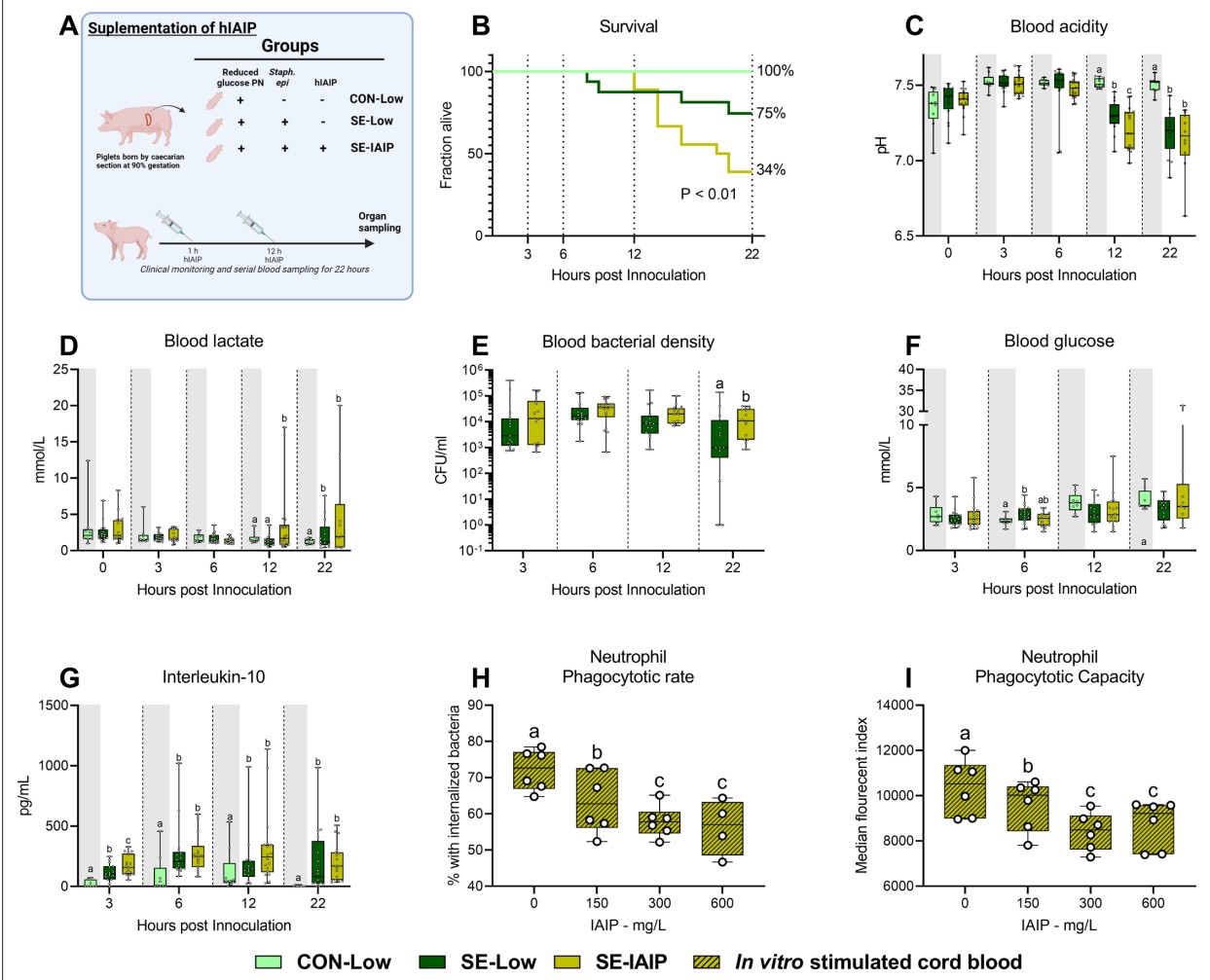

**Figure 7.** Human IAIP intervention, clinical, and immunological results. (**A**) Study overview, preterm pigs were nourished low (5%, 7.2 g/kg/d) glucose parenteral nutrition and infused with either *Staphylococcus epidermidis* or control saline. At one and 12 hr post inoculation, infected animals were treated with either human inter-alpha inhibitor protein (IAIP) (50 mg/kg) or saline and followed for 22 hr while blood samples collected for further analysis. This panel was created with BioRender.com. (**B**) Survival of animals during experiment, presented as time to euthanasia according to predefined humane endpoints with corresponding log-rank test comparing *S. epidermidis* (SE)-Low and SE-IAIP (**C**) Blood glucose, measured by glucose meter 3, 6 and 12 hr after SE inoculation as well as at euthanasia, presented as 95% box plots. (**D, E**) Blood gas parameters collected before inoculation with SE (0 hr) and 3, 6, and 12 hr after as well as at euthanasia, presented as 95% box plots. (**F**) Blood bacterial density in infected groups at 3, 6, 12 and 22 hr after bacterial inoculation, presented as 95% box plots on a logarithmic scale. (**G**) Plasma IL-10 and blood neutrophil fraction at 3, 6, 12 and 22 hr after bacterial inoculation, presented as 95% box plots. (**H, I**) Cord blood neutrophil phagocytic rate and capacity following in vitro challenge with fluorescently labeled *E. coli* and treatment with increasing doses of IAIP, n=7. (**C–H**) Data at each time point analyzed separately, bars labeled with different letters are significantly different from each other (p<0.05), n=8 for control animals, and 10–18 for infected groups. .

The online version of this article includes the following figure supplement(s) for figure 7:

**Figure supplement 1.** Immunological parameters in animals treated with IAIP.

## Discussion

Using a clinically relevant animal model of infections in preterm neonates, we found that a a reduced parenteral glucose supply led to better survival less clinical deterioration, and reduced circulating bacterials, as well as dampened pro-inflammatory responses, all while avoiding hypoglycemia. It was evident that all markers of neonatal sepsis, such as blood pH, lactate, leucocyte/thrombocyte counts were improved by administration of the low glucose PN. At the same time, this nutritional strategy had a clear impact on the hepatic metabolism, reducing the rate of glycolysis while favoring OXPHOS and possibly gluconeogenesis. However, given that most metabolic endpoints are collected at euthanasia we are unsure when the shift in metabolism occurs. Also, to our surprise, we found only

minor changes in the metabolism of circulating leucocytes despite the downregulation of a range of inflammatory genes in the SE-Low group. As such, it is unlikely that the blood leukocyte pool is the crucial link between glucose sensing and immune response to systemic infections. Instead, much of the cytokine production during infections seems to derive from liver macrophages or splenocytes, organs that monitor for blood-borne pathogens (*Minasyan, 2016*; *Heymann and Tacke, 2016*). In the liver transcriptome, there were clear signs of both diminished pro-inflammatory responses and a switch to OXPHOS in SE-Low animals and, when coupled with the plasma metabolome, signs of enhanced gluconeogenesis. We are, however, limited by the use of whole blood/tissue transcriptomics, as opposed to single-cell sequencing, which does not allow for the identification of changes to individual cell types. The observed effects likely occur in separate cell types, with metabolic modulations by glucose supply probably more pronounced in hepatocytes while the immunological changes were altered in the liver immune cells. It is likewise possible that metabolic adaptations in circulating immune cells are present in specific subsets, not detected by the whole blood sequencing. However, from the entirety of our data, the liver appears to be the orchestrator of the immune response to bacterial infection in preterm neonates, and its immune-metabolic response is affected by the circulating glycemic state.

Crucially, a moderately reduced glucose supply did not lead to severe hypoglycemia as we had observed in the previous study (*Muk et al., 2022*). However, we also found that high glucose provision in the early phases of the infection determined the clinical fate of the animals, as those who switched to a reduced glucose regime after the infection was initiated did not have better clinical outcomes than pigs kept on a high-glucose regime throughout the experiment. However, we have not explored the impact of increasing glucose provision after the onset of clinical symptoms, so it is still unknown if the metabolic state of the immune system at the start of the infection determines the outcome. Interestingly though, given that gluconeogenesis seems to be initiated in the SE-Low group during infection. It is possible that preterm newborns have an adequate capacity to utilize other nutrients (i.e. amino acids) to prevent hypoglycemia during glucose-restricted conditions. Therefore, we anticipate that PN with reduced glucose content could be enriched with glucogenic amino acids, which might provide substrates for hepatic gluconeogenesis and help better balance blood sugar levels during neonatal infection.

Reducing glucose intake did not completely protect against the development of sepsis, but combining the low glucose regime with supplementation of hIAIP did further not resolve the inflammatory response and failed to improve survival. A possible underlying cause might be the dose used and its known impact on immune cell activation, which has been reported previously (*Htwe et al., 2018*). We speculate that in our model of neonatal sepsis, exogenously supplemented hIAIP may have negatively altered the responsiveness of neutrophils, thereby potentially increasing pathogen load, resulting in the worsening of the septic condition. Likewise, the premature state of the animals, with already suppressed pro-inflammatory responses (*Nguyen et al., 2016*; *Bæk et al., 2020a*), may have negated the positive results reported elsewhere. Interestingly we observed that porcine IAIP components ITIH1 and bikunin were significantly reduced in circulation only under a high PN glucose regimen. This suggests that a high glucose feeding regimen might be a more suitable, alternative study set up to explore the potential benefits of IAIP supplementation. Further experiments will be needed to optimize the timing and the context of IAIP administration within the current treatment paradigm to uncover its potential treatment benefits in neonatal sepsis.

In conclusion, we identified a clinically relevant regime of reduced glucose provision to prevent sepsis in infected newborn preterm animals, together with relevant mechanistic insights into hepatic immunity and glucose metabolism. Our findings imply that a similar approach for infants with a high risk of infection may help prevent or attenuate the development of life-threatening sepsis. Currently, we are developing a pilot randomized controlled trial to test the feasibility of such a reduced glucose supply in preterm infants with suspected infection, in parallel with more mechanistic pre-clinical studies of modulated hepatic metabolism and systemic immunity in infected preterm animals. Metabolic modulation of immune responses appears to be a promising therapeutic approach for infections in weak and immunocompromised infants.

## Materials and methods

### *S. epidermidis* culture preparation

The full procedure for preparation of the *S. epidermidis* bacteria has been described in detail elsewhere (*Brunse et al., 2018*). In brief, *S. epidermidis* bacteria from frozen stock were cultured in tryptic soy broth overnight whereafter the bacterial density was determined by spectroscopy, and a working solution obtained by diluting with appropriate amounts of sterile saline water.

### Human IAIP preparation

Human IAIP was extracted from fresh frozen human plasma. Starting sample was an intermediate from a typical human plasma fractionation process to purify therapeutically relevant plasma proteins, which was first subjected to solvent/detergent treatment using 1% (v/v) Polysorbate 80 and 0.3% (v/v) Tri-n-butyl phosphate. Next, the sample was purified by three chromatography steps, comprising anion-exchange chromatography on TOYOPEARL GigaCap Q-650M (Tosoh Bioscience) column, followed by two polishing steps using affinity chromatography on a proprietary synthetic chemical ligand support and hydrophobic chromatography on Phenyl PuraBeadHF (both Astrea Bioseparations). Obtained hIAIPs were further concentrated, yielding a preparation with purity >80%.

### In vivo procedures, interventions, and euthanasia

Preterm pigs (Yorkshire × Duroc × Landrace) were delivered by the caesarian section at 90% gestation and rapidly transferred to individual heated, oxygenated incubators. Piglets with prolonged apnea were treated with continuous positive airway pressure until spontaneous respiration was established. Once stable, and still under the influence of maternal anesthesia, all pigs were fitted with an umbilical arterial catheter allowing for the administration of PN and collection of arterial blood samples. Approximately 2 hr after birth, pigs were stratified by birthweight and sex and, in *Exp 1* A, randomly allocated to receive PN with either high (High, 21%, 30 g/kg/d) or reduced (Low, 5%, 7.2 g/kg/d) glucose levels. After initiation of PN, pigs in each group were further randomly allocated to receive either live *S. epidermidis* or control saline (CON) resulting in four groups: SE-High (n=16), SE-Low (n=16), CON-High (n=8), and CON-Low (n=9). For experimental overview see *Figure 1A*. Alongside the glucose intervention, a separate group of preterm pigs were kept on reduced glucose PN, inoculated with *Staph. Epidermidis* and infused with hIAIP (50 mg/kg, SE-IAIP, n=18) one and 12 hr after the start of the experiment. These animals were compared to SE-Low and CON-Low pigs described above separately from the effect of the reduced glucose intervention (see *Figure 7A*).

In a separate follow-up experiment, 41 preterm pigs delivered in the same manner as described above, all started in high glucose PN, were inoculated with *S. epidermidis*, and followed for 12 hr. Pigs were randomly allocated into three groups; with animals either receiving the same high PN glucose supply throughout the whole experiment (SE-12 hr, n=14), or switched to low glucose supply at precisely 3 hr (SE-3 hr, n=15) or 6 hr (SE-6 hr, n=12) post-infection (See *Figure 6A*).

Both experiments were initiated by the administration of *S. epidermidis* ($1 \times 10^9$ CFU/kg, 0 hr) or corresponding volume saline, given as an interatrial infusion over 3 min, whereafter pigs were continuously monitored until 12 or 22 hr post-infection. During experiments, blood samples were collected through the arterial catheter for blood gas, hematology, and cytokine measurements. For blood bacterial enumeration and glucose measurements, blood was collected by jugular vein puncture at the same time points. Blood glucose was measured by capillary puncture using a glucose meter (Accu-Chek, Roche Diagnostics, Denmark). At euthanasia, either scheduled or due to clinical deterioration, animals were sedated and euthanized by intracardial injection of pentobarbital after blood had been collected by an intracardial puncture to yield plasma and serum. Liver samples were then collected and quickly frozen in liquid nitrogen until further analysis.

### Humane endpoints for euthanasia

Across all experiments, animals were continuously monitored for signs of sepsis. These included: changes in respiratory rate, pallor, increased capillary response time, petechial bleeding, pain, bradycardia, and deep lethargy. A blood gas analysis was done on animals showing signs of respiratory or circulatory failure and a pH of less than 7.1 was considered a criterion for immediate euthanasia.

## Inflammatory markers and biochemistry

Plasma samples collected at various time points after bacterial inoculation were analyzed for porcine-specific cytokines using enzyme-linked immunosorbent assay (ELISA), TNF-α (DY690B), IL-10 (DY693B) and IL-6 (DY686, all porcine DuoSet, R&D systems, Abingdon, UK). Plasma samples at planned euthanasia or human endpoints were used for biochemical analysis, using by an Adiva 2120 system (Siemens Healthcare Diagnostics, USA).

## Blood and liver transcriptomics

Total RNA from whole blood collected at 12 hr post-bacterial inoculation and liver at euthanasia was extracted using either MagMAX 96 Blood or Total RNA Isolation Kits (Thermo Fisher, Waltham, MA) for whole transcriptome shotgun sequencing. Briefly, RNA-seq libraries were constructed using 1000 ng RNA and VAHTS mRNA-seq V3 Library Prep Kit for Illumina (Vazyme, China), and 150 bp paired-end reads of the library were generated using the Illumina Hiseq X Ten platform (Illumina, USA). Quality and adapter trimming of raw reads was performed using TrimGalore (Babraham Bioinformatics, UK), and the remaining clean reads (~26 million per sample) were aligned to the porcine genome (Sscrofa11.1) using HISAT2 (*Kim et al., 2019*). The annotated gene information of the porcine genome was obtained from Ensembl (release 99). The script htseq-count (*Anders et al., 2015*) was adopted to generate a gene count matrix, followed by analyses of DEGs using DESeq2 (*Love et al., 2014*).

## Plasma metabolome

Plasma samples at euthanasia or humane endpoints were subjected to proton ($^1$H) NMR spectroscopy-based analysis, as previously described (*Beckonert et al., 2007*). Briefly, samples were mixed with an equal volume of phosphate buffer solution (pH 7.4, 50 mM) prior to the analysis. $^1$H NMR spectra were recorded on a Bruker Avance III 600 MHz NMR spectrometer (Bruker Biospin, Rheinstetten, Germany) equipped with a 5 mm broadband inverse RT (BBI) probe, automated tuning and matching accessory (ATMA), and a cooling unit BCU-05 and an automated sample changer (SampleJet, Bruker Biospin, Rheinstetten, Germany). The spectra were acquired using a standard pulse sequence with water suppression (Bruker pulse program library noesygppr1d), automatically phase- and baseline-corrected using TOPSPIN 3.5 PL6 (Bruker BioSpin, Rheinstetten, Germany), and referenced to the 3-(trimethylsilyl)propionic-2,2,3,3-D4 acid sodium salt (TSP) signal at 0.0 ppm. The final spectra were analyzed using the Signature Mapping software (SigMa) for identification and quantification of metabolites (*Khakimov et al., 2020*).

## Plasma proteome

A subset of plasma samples with sufficient amount from all five timepoints of *Experiment 1* (n=6 from each group, from baseline, 3, 6, 12, and 22 hr post-inoculation) were randomly selected for proteome profiling with shotgun LS-MS/MS-based proteomics. Briefly, plasma was lysed in cold lysis buffer (5% sodium deoxycholate, 50 mM triethyl ammonium bicarbonate, pH 8.5, Sigma-Aldrich) and prepared with filter-aided sample preparation (FASP). Thereafter proteins were recovered using 10 kDa spin filters (VWR), and samples were reduced, alkylated, and digested with trypsin at 37 °C overnight. Peptides were desalted, dried, and stored at –80 °C. At the time of LC-MS/MS analysis, dried peptides were reconstituted in loading buffer (1% acetonitrile and 0.1% formic acid) and determined peptide concentration. Peptides (1 μg) were loaded onto a reversed-phase column on a Thermo Scientific EASY-nLC 1200 nano-liquid chromatography system, connected to a Thermo Scientific Q Exactive HF-X mass spectrometer equipped with a Nanospray Flex ion source. A hybrid spectral library was created from DDA (data-dependent analysis) and DIA (data-independent analysis) search results. For DDA, a modified TOP15 method was adopted to cover the full range MS1 scan with m/z range of 330–1650, followed by 15 data dependent MS2 scans (*Kelstrup et al., 2018*). Data were searched using SpectroMine software (Biognosys, version 2.8), using the Uniprot *Sus scrofa* library from UniProt with the set false discovery rate (FDR) of 1% and maximal missed cleavages of 2. For DIA, the method consisted of one full range MS1 scan and 21 DIA segments to search peptides using Spectronaut software with FDR set to 1%, as previously described (*Bruderer et al., 2017*).

## Effect of human IAIP on in vitro neutrophil function

In addition to the animal study, an in vitro experiment to investigate the effect of hIAIP on the phagocytic function of neutrophils was performed. Whole cord blood samples from preterm pigs, delivered

in the same manner as described above (n=8), were divided and treated with increasing levels of hIAIP (0–600 mg/L, mimicking normal plasma levels in termborn infants *Baek et al., 2003*) and a standard dose of fluorescently marked *Escherichia coli* (pH Rhodo, Thermo Fisher) for 30 min at 37 °C. Samples were then washed, and run on a flow cytometer and the neutrophil population was identified, as described in detail elsewhere (*Bæk et al., 2020b*). Neutrophil phagocytic rate was defined as the fraction of neutrophils with internalized bacteria and the phagocytic capacity as their median fluorescent intensity.

## Statistics

Data on survival and clinical outcomes were analyzed using Stata 14 (StataCorp, USA). Survival curves were compared using the log-rank test while data collected during the studies were analyzed separately at each timepoint with a two-way ANOVA, with litter and sex as covariates, hereafter group differences were identified with post hoc testing using Tukey's test. If necessary, data were logarithmically transformed to obtain normal distribution; data that could not conform to normality were analyzed by a non-parametric test.

Omics data were analysed using R Studio 4.1.2 (R Studio, Boston, MA). Metabolome and proteome data were analysed by a linear mixed-effect model with treatment as a fixed factor and litter as a random factor at each time point, followed by Tukey Post-hoc pair-wise comparisons using *lme4* and *multcomp* packages (*Bates et al., 2014*). For transcriptomics, significant DEGs among groups were identified by DESeq2 using Benjamin-Hochberg (BH)-adjusted p-value <0.1 as the cut-off. To control type I error, p values were further adjusted by FDR ($\alpha$=0.1) into q values (*Pollard et al., 2021*). An FDR-adjusted p-value <0.1 for blood and FDR-adjusted p-value <0.05 for liver were regarded as statistically significant. Representative plots of single protein or metabolite were presented as violin dot plots with median and interquartile ranges. All reported measures were evaluated for normal distribution, and logarithmic transformation was performed if necessary.

Pathway enrichment analysis for metabolome data is generated with MetaboAnalyst 5.0 (*Pang et al., 2022*) and illustrated with BioRender. For proteomics, gene ontology (GO) and KEGG pathway enrichment analyses for DEPs were performed using DAVID (*Huang et al., 2009*) and a BH-adjusted p-value <0.05 was considered statistically significant. For transcriptomics, gene set enrichment analysis (GSEA) for DEGs was performed using the *fgsea* package with R and GSEA 4.2.2 (UC San Diego and Broad Institute) using GO, KEGG, and Hallmark database, and pathways with adjusted p-value <0.05 were considered statistically significant (*Subramanian et al., 2005*). Oimcs datasets were scaled to unit variance prior to PCA. PCA score plots were generated and the number of principal components was determined by cross-validation using R *pcaMethods and prcomp* packages and plotted with the ggplot2 package, and heatmaps were generated using R package *pheatmap*.

## Study approval

The animal studies and experimental procedures were approved by the Danish Animal Experiments Inspectorate (license no. 2020-15-0201-00520), which complies with the EU Directive 2010/63 (legislation for the use of animals in research).

## Acknowledgements

The authors thank Thomas Thymann, Simone Offersen, Karoline Aasmul-Olsen, Malene Spiegelhauer, Kristina Larsen, Britta Karlsson, and Jane C Povlsen for the assistance in animal experiments and omic data acquirement and Lucia Gnauer for provision of the purified human IAIP material.

## Additional information

### Competing interests

Alessandra Maria Casano, Bagirath Gangadharan, Ivan Bilic: employed by Takeda Pharmaceuticals at the the time of the study, which funded part of the experiment. The other authors declare that no competing interests exist.

### Funding

| Funder | Grant reference number | Author |
| --- | --- | --- |
| Novo Nordisk Fonden | NNF22OC0078747 | Duc Ninh Nguyen |
| Takeda Pharmaceuticals International | | Duc Ninh Nguyen Per Torp Sangild |

The funders had no role in study design, data collection and interpretation, or the decision to submit the work for publication.

### Author contributions

Ole Bæk, Conceptualization, Data curation, Formal analysis, Investigation, Visualization, Methodology, Writing – original draft, Writing – review and editing; Tik Muk, Investigation, Visualization, Methodology, Writing – review and editing; Ziyuan Wu, Investigation, Writing – review and editing; Yongxin Ye, Bekzod Khakimov, Alessandra Maria Casano, Bagirath Gangadharan, Anders Brunse, Methodology, Writing – review and editing; Ivan Bilic, Conceptualization, Methodology, Writing – review and editing; Per Torp Sangild, Conceptualization, Funding acquisition, Methodology, Project administration, Supervision, Writing – review and editing; Duc Ninh Nguyen, Conceptualization, Supervision, Investigation, Methodology, Project administration, Writing – review and editing

### Author ORCIDs

Ole Bæk ⓘ https://orcid.org/0000-0003-0598-0952
Duc Ninh Nguyen ⓘ https://orcid.org/0000-0002-4997-555X

### Ethics

The animal studies and experimental procedures were approved by the Danish Animal Experiments Inspectorate (license no. 2020-15-0201-00520), which complies with the EU Directive 2010/63 (legislation for the use of animals in research).

Reviewer #1 (Public review): https://doi.org/10.7554/eLife.97830.3.sa1
Reviewer #2 (Public review): https://doi.org/10.7554/eLife.97830.3.sa2
Author response https://doi.org/10.7554/eLife.97830.3.sa3

---

## Additional files

### Supplementary files

Supplementary file 1. Differentially expressed genes from hepatic transcriptome.

Supplementary file 2. Differentially expressed genes in whole blood transcriptome.

Supplementary file 3. Differentially expressed metabolites in plasma metabolome.

Supplementary file 4. Differentially expressed proteins in plasma proteome.

MDAR checklist

Source data 1. Clinical and immunological data for main experiment, related to *Figure 1* and *Figure 7*.

### Data availability

All data generated or analysed during this study are included in the manuscript and supporting files. Source data files have been provided for *Figures 1–7*.

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
