## [Editor Report · eLife Assessment]

The study follows up on previous work suggesting that lower glucose concentrations are protective from sepsis but put the patient at risk for hypoglycemia. In this paper, the authors identify that a slightly higher dose of glucose is still protective but no longer puts the patients at risk for hypoglycemia. The study is **important**, supported by **convincing** data, and will be of interest to a broad audience.

---

## [Referee Report · Reviewer #1 (Public review)]

Summary:

In this manuscript the authors follow up on their published observation that providing a lower glucose parental nutrition (PN) reduces sepsis from a common pathogen [Staphylococcus epidermitis (SE)] in preterm piglets. Here they found that a slightly higher dose of glucose could thread the needle and get the protective effects of low glucose without incurring significant hypoglycemia. They then investigate whether change in low glucose PN impacts metabolism to confer this benefit. The finding that lower glucose reduces sepsis is important as sepsis is a major cause of morbidity and mortality in preterm infants, and adjusting PN composition is a feasible intervention.

Strengths:

(1) They address a highly significant problem of neonatal sepsis in preterm infants using a preterm piglet model.

(2) They have compelling data in this paper (and in a previous publication, ref 27) that low glucose PN confers a survival advantage. A downside of the low glucose PN is hypoglycemia which they mitigate in this paper by using a slightly high amount of glucose in the PN.

(3) The experiment where they change PN from high to low glucose after infection is very important to determine if this approach might be used clinically. Unfortunately, this did not show an ability to reduce sepsis risk with this approach.

(4) They produce an impressive multiomics data set from this model of preterm piglet sepsis which is likely to provide additional insights into the pathogenesis of preterm neonatal sepsis.

Weaknesses:

(1) Piglets on the low glucose PN had consistently lower density of SE (~1 log) across all timepoints. This may be due to changes in immune response leading to better clearance or it could be due to slower growth in lower glucose environment. These possibilities are not fully disentangled in this study.

(2) Many differences in the different omics (transcriptomics, metabolomics, proteomics) were identified in the SE-LOW vs SE-HIGH comparison. Since the bacterial load is very different between these conditions, could the changes be due to bacterial load rather than metabolic reprograming from the low glucose PN? The authors argue in supplementary figure 1F that density of SE in blood does not correlate with sepsis implying that bacterial load is not the driver of outcome. The authors recently published some additional analysis that may be helpful to reference in this manuscript.

(3) Further, expanding upon a model to better understand the complex relationship between differences in supplemental glucose infusion, blood glucose levels, bacterial load, host responses and how they impact the development of sepsis would be helpful. These complex relationships are difficult to fully disentangle, but one could consider infusing the same quantity of heat-killed bacteria under different glucose conditions to see if the glucose levels drive outcomes independently of bacterial burden.

---

## [Referee Report · Reviewer #2 (Public review)]

The authors demonstrate that a low parenteral glucose regimen can lead to improved bacterial clearance and survival from Staph epi sepsis in newborn pigs without inducing hypoglycemia, as compared to a high glucose regimen. Using RNA-seq, metabolomic, and proteomic data, the authors conclude that this is primarily mediated by altered hepatic metabolism.

The authors have addressed the concerns raised by the reviewers in their revised manuscript and have added additional information in the results and discussion part.

Please address in Fig. 3- the genes PGM2 and GCK, which the authors mention, are downregulated in SE-Low compared to SE-high, but these are actually less downregulated in the SE group compared to Control group, where the the Con-low shows even more decrease in these genes compared to Con-high. So if anything, these genes are getting upregulated by infection.

---

## [Author Response]

The following is the authors’ response to the original reviews.

**Public Reviews:**

**Reviewer #1 (Public Review):**
Summary:In this manuscript, the authors follow up on their published observation that providing a lower glucose parental nutrition (PN) reduces sepsis from a common pathogen [Staphylococcus epidermitis (SE)] in preterm piglets. Here they found that a higher dose of glucose could thread the needle and get the protective effects of low glucose without incurring significant hypoglycemia. They then investigate whether the change in low glucose PN impacts metabolism to confer this benefit. The finding that lower glucose reduces sepsis is important as sepsis is a major cause of morbidity and mortality in preterm infants, and adjusting PN composition is a feasible intervention.Strengths:(1) They address a highly significant problem of neonatal sepsis in preterm infants using a preterm piglet model.(2) They have compelling data in this paper (and in a previous publication, ref 27) that low glucose PN confers a survival advantage. A downside of the low glucose PN is hypoglycemia which they mitigate in this paper by using a slightly high amount of glucose in the PN.(3) The experiment where they change PN from high to low glucose after infection is very important to determine if this approach might be used clinically. Unfortunately, this did not show an ability to reduce sepsis risk with this approach. Perhaps this is due to the much lower mortality in the high glucose group (~20% vs 87% in the first figure).(4) They produce an impressive multiomics data set from this model of preterm piglet sepsis which is likely to provide additional insights into the pathogenesis of preterm neonatal sepsis.Weaknesses:(1) The high glucose control gives very high blood glucose levels (Figure 1C). Is this the best control for typical PN and glucose control in preterm neonates? Is the finding that low glucose is protective or high glucose is a risk factor for sepsis?

This work is a follow-up from our previous work where we explored different PN glucose regimens. Taken together our experiments heavily imply that glucose provision is associated to severity in a seemingly linear manner. In the clinical setting, there is no fixed glucose provision, but guidelines specify ranges that are acceptable. However, these guidelines do not take possible infections into account and are designed to optimize growth outcomes. Increased provision of glucose to preterm neonates may therefore increase their infection risk, but parenteral glucose cannot be entirely avoided as it would lead to hypoglycaemia and associated brain damage. In the present paper the reduced glucose PN reflects the lowest end of the recommended PN glucose intake. More work is needed to figure out the best glucose provision to infected preterm newborns, balancing positive and negative factors.

(2) In Figure 1B, preterm piglets provided the high glucose PN have 13% survival while preterm piglets on the same nutrition in Figure 6B have ~80% survival. Were the conditions indeed the same? If so, this indicates a large amount of variation in the outcome of this model from experiment to experiment.

In the follow-up experiment outlined in Figure 6 we reduced the follow-up time to 12 hours in an effort to minimize the suffering of the animals. We did this because we could detect relevant differences in the immune response between High and low glucose infected pigs as 12 hours. If we had extended the follow-up experiment to 22 hours we would likely have seen a much increased mortality.

(3) Piglets on the low glucose PN had consistently lower density of SE (~1 log) across all time points. This may be due to changes in immune response leading to better clearance or it could be due to slower growth in a lower glucose environment.

We agree with this assessment and have adjusted our result section to reflect this.

(4) Many differences in the different omics (transcriptomics, metabolomics, proteomics) were identified in the SE-LOW vs SE-HIGH comparison. Since the bacterial load is very different between these conditions, could the changes be due to bacterial load rather than metabolic reprogramming from the low glucose PN?

We analyzed the relationship between bacterial burdens and mortality and found that it did not correlate within each of the treatment groups. We have now added this data to the results section as supplemental and report this fact in the section called “Reduced glucose supply increases hepatic OXPHOS and gluconeogenesis and attenuates inflammatory pathways”. This finding inspired us to further explore the relationship between bacterial burdens and infection responses in our model which has resulted in our recent preprint: Wu et at. Regulation of host metabolism and defense strategies to survive neonatal infection. BioRxiv 2024.02.23.581534; doi: https://doi.org/10.1101/2024.02.23.581534

**Reviewer #2 (Public Review):**
Summary:The authors demonstrate that a low parenteral glucose regimen can lead to improved bacterial clearance and survival from Staph epi sepsis in newborn pigs without inducing hypoglycemia, as compared to a high glucose regimen. Using RNA-seq, metabolomic, and proteomic data, the authors conclude that this is primarily mediated by altered hepatic metabolism.Strengths:Well-defined controls for every time point, with multiple time points and biological replicates. The authors used different experimental strategies to arrive at the same conclusion, which lends credibility to their findings. The authors have published the negative findings associated with their study, including the inability to reverse sepsis-related mortality after switching from SE-high to SE-low at 3h or 6h and after administration of hIAIP.Weaknesses:(1) The authors mention, and it is well-known, that Staph epi is primarily involved in late-onset sepsis. The model of S. epi sepsis used in this study clearly replicates early-onset sepsis, but S. epi is extremely rare in this time period. How do the authors justify the clinical relevance of this model?

The distinction between early and late onset sepsis makes sense clinically because they are likely to be caused by different organisms and therefore require different empirical antibiotic regimes. Early onset sepsis is caused by organisms transferred perinatally often following chorioamnionitis or uro-gential maternal infections (Strep. agalacticae/*E. coli*) whereas Late onset sepsis is likely caused by organisms from indwelling catheters or mucosal surfaces, most often coagulase negative staphylococci. Timing of an infection after birth of course plays a role, but the virulence factors of the pathogen probably plays a large role in shaping the immune response. Therefore, even though the infection in our model is initiated on the first day after birth, the organism that we use, *Staph epidermidids*, makes it a better model for pathogenesis of late onset sepsis. However, it is also important to acknowledge that the pathophysiology of “sepsis” may be similar despite timing and pathogen and depends on the degree of immune activation and downstream effects on organs.

(2) The authors find that the neutrophil subset of the leukocyte population is diminished significantly in the SE-low and SE-high populations. However, they conclude on page 10 that "modulations of hepatic, but not circulating immune cell metabolism, by reduced glucose supply..." and this is possible because the authors have looked at the entire leukocyte transcriptome. I am curious about why the authors did not sequence the neutrophil-specific transcriptome.

We collected the whole blood transcript during the experiments, which reflect the transcription profile of all the circulating leucocytes. Since we did not do single cell RNA sequencing during the experiment there is no possibility of isolating the neutrophil transcriptome at this time. Your point however is valid and we will reconsider incorporating single cell transcriptomics in future experiments.

(3) The authors use high (30g/k/d) and low (7.2g/k/d) glucose regimens. These translate into a GIR of 21 and 5 mg/k/min respectively. A normal GIR for a preterm infant is usually 5-8, and sometimes up to 10. Do the authors have a "safe GIR" or a threshold they think we cannot cross? Maybe a point where the metabolism switch takes place? They do not comment on this, especially as GIR and glucose levels are continuous variables and not categorical.

Our reduced glucose PN was chosen as it corresponded with the low end of recommended guidelines for PN glucose intake. There likely is not a “safe GIR” as the clinical responses to glucose intake during infections do not seem binary but increase with glucose intake. It is also important to remember that the reduced glucose intervention still resulted in significant morbidity and a 25% mortality within 22 hours. There is therefore still vast room for improvement, but even though further reduction in PN glucose would probably provide further protection it would entail dangerous hypoglycaemia (as described in our previous paper). The findings in this current paper has prompted us to explore several strategies to replace parenteral glucose with alternative macronutrients. Thus, the optimal PN for infected newborns would probably differ from standard PN in all macronutrients and will require much more pre- and clinical research.

(4) In Figures 2B and C the authors show that SE-high and SE-low animals have differences in the oxphos, TCA, and glycolytic pathways. The authors themselves comment in the Supplementary Table S1B, E-F that these same metabolic pathways are also different in the Con-Low and Con-high animals, it is just the inflammatory pathways that are not different in the non-infected animals. How can they then justify that it is these metabolic pathways specifically which lead to altered inflammatory pathways, and not just the presence of infection along with some other unfound mechanism?

It is to be expected that the inflammatory pathways do not differ between the Con-Low and Con-High groups as there is no infection to induce these pathways. The identified metabolic pathways that differ between SE-High and SE-Low animals seem to us the best explanation of the differences in clinical phenotype.

(5) The authors mention in Figure 1F that SE-low animals had lower bacterial burdens than SE-high animals, but then go on to infer that the inflammatory cytokine differences are attributed to a rewiring of the immune response. However, they have not normalized the cytokine levels to the bacterial loads, as the differences in the cytokines might be attributed purely to a difference in bacterial proliferation/clearing.

Please see our response to reviewer #1

(6) The authors mention that switching from SE-high to SE-low at 3 or 6 h time points does not reduce mortality. Have the authors considered the reverse? Does hyperglycemia after euglycemia initially, worsen mortality? That would really conclude that there is some metabolic reprogramming happening at the very onset of sepsis and it is a lost battle after that.

A very good point that we have not explored yet, we have added this consideration to the discussion and slightly amended our conclusions of this follow-up experiment.

**Reviewer #3 (Public Review):**
Summary:Baek and colleagues present important follow-up work on the role of serum glucose in the management of neonatal sepsis. The authors previously showed high glucose administration exacerbated neonatal sepsis, while strict glucose control improved outcomes but caused hypoglycemia. In the current report they examined the effect of a more tailored glucose management approach on outcomes and examined hepatic gene expression, plasma metabolome/proteome, blood transcriptome, as well as the the therapeutic impact of hIAIP. The authors leverage multiple powerful approaches to provide robust descriptive accounts of the physiologic changes that occur with this model of sepsis in these various conditions. Strengths:(1) Use of preterm piglet model.(2) Robust, multi-pronged approach to address both hepatic and systemic implications of sepsis and glucose management.(3) Trial of therapeutic intervention - glucose management (Figure 6), hIAIP (Figure 7).Weaknesses:(1) The translational role of the model is in question. CONS is rarely if ever a cause of EOS in preterm neonates. The model. uses preterm pigs exposed at 2 hours of age. This model most likely replicates EOS.

Please see our response to Reviewer #2

(2) Throughout the manuscript it is difficult to tell from which animals the data are derived. Given the ~90% mortality in the experimental CONS group, and 25% mortality in the intervention group, how are the data from animals "at euthanasia" considered? Meaning - are data from survivors and those euthanized grouped together? This should be clarified as biologically these may be very different populations (ie, natural survivor vs death).

This is a very valid point. For all endpoints that are analyzed “at euthanasia” the age of the animal will vary. Some will have been euthanized early due to clinical deterioration and some will have survived all the way to the end of the experiment. This needs to be kept in mind when interpreting the results. We have further highlighted this point in the discussion and made it clear to the reader at what time-point each analysis was performed.

(3) With limited time points (at euthanasia) for hepatic transcriptomics (Figure 2), plasma metabolite (Figure 3) blood transcriptome (Figure 4), and plasma proteome (Figure 5) it is difficult to make conclusions regarding mechanisms preceding euthanasia. Per methods, animals were euthanized with acidosis or clinical decompensation. Are the reported findings demonstrative of end-organ failure and deterioration leading to death, or reflective of events prior?

Yes, all organ specific endpoints are snapshots of the state of the animals at the time of euthanasia, pooling together animals that succumbed to sepsis and those that survived to 22 hours post infection. These results therefore reflect the end-state of the infection we cannot be sure when the differences between groups manifested themselves. However, given the stark differences in plasma lactate at 12 hours post infection it is likely that changes to metabolism occurred before most of animals succumbed to sepsis.

We agree this is a weakness in our model, but we have since published a pre-print where we have further explored how metabolic adaptations shape the fate of similarly infected preterm pigs: BioRxiv 2024.02.23.581534; doi: https://doi.org/10.1101/2024.02.23.581534

(4) Data are descriptive without corresponding "omics" from interventions (glucose management and/or hIAIP) or at least targeted assessment of key differences.

We only did in-depth analysis of the glucose intervention as this showed the most promising clinical effects that warranted further in-depth investigation. It is possible that further insights could be gained from in-depth analysis of the other interventions but given that there were no obvious clinical befits we refrained from that.

**Recommendations for the authors:**

**Reviewer #1 (Recommendations For The Authors):**
I am intrigued that mortality was not correlated to bacterial burden. Please provide the "data not shown" as this would help the reader understand better whether the difference in bacterial burden is driving the phenotypes and findings of the low glucose group.

We have added this data to supplementary figure 1.

**Reviewer #2 (Recommendations For The Authors):**
(1) I would urge the authors to consider a neutrophil-specific transcriptomic analysis. I understand that this would add significantly to the resubmission process. If the authors wish to include that as a future direction instead, they need to specifically mention the limitations of whole blood transcriptomics and how different immune cell types react differently to bacterial antigens.

We agree with your considerations but we cannot include that data using the whole blood method applied in the experiment. We have added your consideration to the discussions.

(2) I urge the authors to remove any impression that this is a model of late-onset sepsis, which is implied from the introduction, lines 3 and 4.

Our intention was not to directly suggest that our model is a perfect reflection of late-onset sepsis but rather to highlight the relevance of using a pathogen commonly associated with LOS. We believe our model primarily captures the effects of intense pro-inflammatory immune activation, which may have parallels with various forms of sepsis, including LOS.

**Reviewer #3 (Recommendations For The Authors):**
Drawing on the robust nature of your "omics", identify key measures and test whether they are altered earlier in the development of clinical sepsis. Test whether these are altered by the intervention.

A very valid point, at the moment it is not possible for us to explore this within the confines of these experiments. But, building upon these findings and the ones in our recent preprint we are confident that shifts in hepatic ratio of Oxidative phosphorylation and gluconeogenesis vs glycolysis shape the immune response to infections in neonates. In our upcoming experiments we are planning to incorporate plasma metabolomics at earlier timepoints to monitor when shifts in metabolism occur. However, given the heterogeneity of pigs, as opposed to inbred rodent models, sacrificing animals at fixed timepoints to gauge their organ function will be hard to interpret as it is impossible to know what the end state of the particular animal would have been. Therefore longitudinal sampling of liver tissue, during the course of infection would be challenging.